# Small molecule inhibitors reveal an indispensable scaffolding role of RIPK2 in NOD2 signaling

Matous Hrdinka[1,†,‡] (ID), Lisa Schlicher[1,†], Bing Dai[2], Daniel M Pinkas[3], Joshua C Bufton[3,§], Sarah Picaud[3], Jennifer A Ward[3,4], Catherine Rogers[3,4], Chalada Suebsuwong[5,¶], Sameer Nikhar[6], Gregory D Cuny[6], Kilian VM Huber[3,4] (ID), Panagis Filippakopoulos[3], Alex N Bullock[3], Alexei Degterev[2,*] (ID) & Mads Gyrd-Hansen[1,**] (ID)

## Abstract

RIPK2 mediates inflammatory signaling by the bacteria-sensing receptors NOD1 and NOD2. Kinase inhibitors targeting RIPK2 are a proposed strategy to ameliorate NOD-mediated pathologies. Here, we reveal that RIPK2 kinase activity is dispensable for NOD2 inflammatory signaling and show that RIPK2 inhibitors function instead by antagonizing XIAP-binding and XIAP-mediated ubiquitination of RIPK2. We map the XIAP binding site on RIPK2 to the loop between β2 and β3 of the N-lobe of the kinase, which is in close proximity to the ATP-binding pocket. Through characterization of a new series of ATP pocket-binding RIPK2 inhibitors, we identify the molecular features that determine their inhibition of both the RIPK2-XIAP interaction, and of cellular and *in vivo* NOD2 signaling. Our study exemplifies how targeting of the ATP-binding pocket in RIPK2 can be exploited to interfere with the RIPK2-XIAP interaction for modulation of NOD signaling.

**Keywords** kinase inhibitor; NOD2 signaling; RIPK2; ubiquitin; XIAP
**Subject Categories** Autophagy & Cell Death; Immunology
**The EMBO Journal (2018) 37: e99372**

## Introduction

Receptor-interacting kinases (RIPKs) are components of innate immune receptor signaling complexes where they become ubiquitinated and contribute to NF-κB-mediated inflammatory signaling and cell death (Hrdinka & Gyrd-Hansen, 2017; Annibaldi & Meier, 2018). The intracellular bacteria-sensing receptors NOD1 and NOD2 (nucleotide-oligomerization domain-containing proteins 1 and 2) stimulate inflammatory signaling by promoting RIPK2 ubiquitination in response to binding of iE-DAP (D-glutamyl-meso-diaminopimelic acid) and MDP (muramyl dipeptide) constituents of bacterial peptidoglycan, respectively (Girardin *et al*, 2003). NOD1/2 signaling contributes to gastro-intestinal immunity (Philpott *et al*, 2014), and genetic variants in NOD2 are the strongest susceptibility factors to Crohn's disease—one of the two major inflammatory bowel diseases afflicting millions in Europe and North America alone (Hugot *et al*, 2001; Ogura *et al*, 2001; Ananthakrishnan, 2015). Mutations of NOD2 have also been implicated in other auto-inflammatory granulomatous pathologies such as Blau's syndrome and early-onset sarcoidosis (Caso *et al*, 2015).

Stimulation of NOD2 recruits RIPK2 along with several ubiquitin (Ub) ligases, including IAP (Inhibitor of Apoptosis) proteins and LUBAC (linear ubiquitin chain assembly complex) (Hasegawa *et al*, 2008; Bertrand *et al*, 2009; Tao *et al*, 2009; Damgaard *et al*, 2012; Yang *et al*, 2013; Watanabe *et al*, 2014). These ligases, together with deubiquitinases, coordinate the conjugation of Lys63- and Met1-linked Ub chains (Lys63-Ub and Met1-Ub) on RIPK2 to facilitate signal transduction (Hitotsumatsu *et al*, 2008; Fiil *et al*, 2013; Draber *et al*, 2015; Hrdinka *et al*, 2016). Lys63-Ub and Met1-Ub are central for productive innate immune signaling and transcription of nuclear factor-κB (NF-κB) target genes (Hrdinka & Gyrd-Hansen, 2017). Lys63-Ub is recognized by the TAK1-TAB 2/3 (TGFβ-activated kinase 1; TAK1-binding protein 2/3) kinase complex, and Met1-Ub is bound by the IKK (IκB kinase) complex through the subunit NEMO (NF-κB essential modifier; also known as IKKγ). In

1    Nuffield Department of Clinical Medicine, Ludwig Institute for Cancer Research, University of Oxford, Oxford, UK
2    Department of Developmental, Molecular & Chemical Biology, Tufts University School of Medicine, Boston, MA, USA
3    Nuffield Department of Clinical Medicine, Structural Genomics Consortium, University of Oxford, Oxford, UK
4    Nuffield Department of Clinical Medicine, Target Discovery Institute, University of Oxford, Oxford, UK
5    Department of Chemistry, University of Houston, Houston, TX, USA
6    Department of Pharmacological and Pharmaceutical Sciences, University of Houston, Houston, TX, USA
    *Corresponding author. Tel: +1 617 636 0491; E-mail: alexei.degterev@tufts.edu
    **Corresponding author. Tel: +44 (0) 1865 617508; E-mail: mads.gyrd-hansen@ludwig.ox.ac.uk
    †These authors contributed equally to this work
    ‡Present address: Department of Haematooncology, University Hospital Ostrava, Ostrava-Poruba, Czech Republic
    §Present address: Department of Biochemistry, University of Bristol, Bristol, UK
    ¶Present address: Department of Pharmacological Sciences, Icahn School of Medicine at Mount Sinai, New York, NY, USA

turn, the kinase complexes are activated, leading to phosphorylation, ubiquitination, and degradation of the NF-κB inhibitory factor IκBα and activation of MAP kinases (Hrdinka & Gyrd-Hansen, 2017).

XIAP (X-linked IAP) is indispensable for NOD2 signaling and familial mutations in XIAP that impact on its function cause severe immunodeficiency with variable clinical presentation, including early-onset chronic colitis in ~20% of afflicted individuals (Bauler et al, 2008; Krieg et al, 2009; Damgaard et al, 2012, 2013; Speckmann et al, 2013; Pedersen et al, 2014). The ubiquitination of RIPK2 by XIAP facilitates recruitment of LUBAC (Damgaard et al, 2012), which in turn conjugates Met1-Ub on RIPK2 (Fiil et al, 2013). Previous data using small molecule inhibitors suggested that catalytic activity of RIPK2 may contribute to XIAP-mediated RIPK2 ubiquitination (Canning et al, 2015; Nachbur et al, 2015). Consequently, the activity of RIPK2 has been implicated in a subset of systemic granulomatous inflammatory diseases (Jun et al, 2013) and, in particular, ablation of Ripk2 or inhibition of RIPK2 by small-molecule kinase inhibitors showed benefits in mouse models of multiple sclerosis (Shaw et al, 2011; Nachbur et al, 2015) and Crohn's disease-like ileitis (Tigno-Aranjuez et al, 2014), positioning RIPK2 as a new target against human inflammatory diseases. However, the molecular basis for the cross-talk between the kinase activity of RIPK2 and its role as a critical ubiquitinated scaffold downstream of NOD1/2 remains enigmatic.

Here, we reveal that RIPK2 kinase activity is dispensable for NOD2 inflammatory signaling, show that RIPK2 inhibitors function instead by antagonizing XIAP-binding and XIAP-mediated ubiquitination of RIPK2, and identify structural features of RIPK2 required for XIAP binding and for the design of efficient small molecule inhibitors of NOD1/2-RIPK2-dependent signaling. Our study exemplifies how targeting the ATP-binding pocket in RIPK2 can be exploited to interfere with the RIPK2-XIAP interaction for modulation of NOD signaling.

## Results

### RIPK2 kinase activity is dispensable for NOD2 signaling

Previous reports showed that tyrosine-kinase inhibitors such as ponatinib, gefitinib, and the RIPK2-selective kinase inhibitors GSK583 and WEHI-345 inhibit cellular responses to the NOD2 agonist MDP (or L18-MDP) by antagonizing RIPK2 function (Tigno-Aranjuez et al, 2010; Canning et al, 2015; Nachbur et al, 2015; Haile et al, 2016). In concordance, ponatinib and GSK583 inhibited the degradation of IκBα and NF-κB-mediated production of the chemokine CXCL8 in a dose-dependent manner in U2OS/NOD2 cells stimulated with L18-MDP (Fig 1A–C). Of note, the U2OS/NOD2 cells used in this study express doxycycline (DOX)-inducible HA-NOD2 and respond to L18-MDP without addition of DOX due to leakiness of the promoter (Fiil et al, 2013). Small molecule kinase inhibitors are categorized into multiple classes, depending on their mode of binding (Roskoski, 2016). This includes type I inhibitors that interact exclusively within the ATP-binding pocket, type II inhibitors that bind both to the ATP, and an additional back pocket created when the activation segment of a kinase adopts an inactive conformation, and type III molecules that bind exclusively to this allosteric back

pocket. Curiously, we observed that a subset of known RIPK2 inhibitors belonging to different classes displayed potent (nanomolar) cellular activities, including ponatinib (a type II inhibitor) and GSK583 (an ATP-competitive type I inhibitor), and that these molecules also antagonized NOD2-mediated ubiquitination of RIPK2 (Figs 1C and EV1A; Canning et al, 2015). This implied that the kinase activity of RIPK2 is required for its ubiquitination and, thus, for NOD2 responses. To directly investigate this, we first ablated RIPK2 (RIPK2 KO) by CRISPR-mediated gene editing in U2OS/NOD2 cells (Fig EV1B and C). As expected, degradation of IκBα and production of CXCL8 in response to L18-MDP were completely inhibited in RIPK2 KO cells (Park et al, 2007; Fig 1D and E). Reintroduction of wild-type (WT) RIPK2 restored RIPK2 ubiquitination and CXCL8 production, and partially restored IκBα degradation, confirming that the signaling defect was due to the absence of RIPK2 (Fig 1D and E). Next, RIPK2 KO cells were reconstituted with kinase-dead human RIPK2 variants in which the ATP-binding lysine 47 was substituted for arginine (K47R) or the catalytic aspartate 146 in the "HRD" motif (HHD in human RIPK2) was substituted for asparagine (D146N) (Pellegrini et al, 2017; Figs 1F and G, and EV1D). Strikingly, introduction of both kinase-dead RIPK2 mutants restored NOD2 signaling and CXCL8 production to a similar level as with WT RIPK2 in two independent RIPK2 KO clones, showing that the catalytic function is not needed for RIPK2's role in NOD2-dependent inflammatory signaling (Figs 1G and H, and EV1E).

RIPK2 ubiquitination in response to L18-MDP was also not affected by the K47R and D146N mutations, which is surprising since kinase inhibitors blocked RIPK2 ubiquitination (Figs 1C and 2A). Moreover, ponatinib prevented ubiquitination of the kinase-dead RIPK2 variants after NOD2 stimulation and antagonized their capacity to induce NF-κB activation (Fig 2A and B), suggesting that the inhibition of RIPK2 ubiquitination and NOD2 signaling by ponatinib is independent of its inhibition of RIPK2 kinase activity. Although RIPK2 is a high-affinity cellular target of ponatinib, the molecule is a promiscuous kinase inhibitor (Fauster et al, 2015; Najjar et al, 2015; Appendix Fig S1A; Dataset EV1). To determine whether ponatinib's inhibitory activity was a result of its binding to RIPK2, we substituted the threonine 95 "gatekeeper" residue with a bulky tryptophan (T95W) to prevent ponatinib's binding to the RIPK2 ATP-binding pocket (Fig 2C). We first confirmed that the T95W mutation indeed ablated the binding of ponatinib to RIPK2 in cells at the concentrations used to inhibit signaling using a nano-bioluminescence resonance energy transfer (nanoBRET) assay (Vasta et al, 2018) and cellular thermal shift assay (CETSA; Jafari et al, 2014), which measure cellular target engagement. Use of our recently reported ponatinib-derived kinase tracer SGC-590001 (Vasta et al, 2018) in conjunction with nanoLuc-RIPK2 showed that the T95W mutation prevented detectable interaction at inhibitor concentrations up to more than 100 nM (Fig 2D). In accordance with this and with previous findings (Canning et al, 2015), ponatinib (300 nM) induced a substantial thermal shift indicative of binding to WT RIPK2 (and K47R RIPK2), but this was not observed for the RIPK2 T95W and K47R+T95W mutants (Appendix Fig S1B). Introduction of the T95W mutation, either alone or in combination with kinase-dead RIPK2 substitutions, largely abolished the inhibitory effect of ponatinib on RIPK2-induced NF-κB activation (Fig 2B) and on NOD2 signaling as determined by RIPK2 ubiquitination, IκBα degradation, phosphorylation of NF-κB (p65), and production of

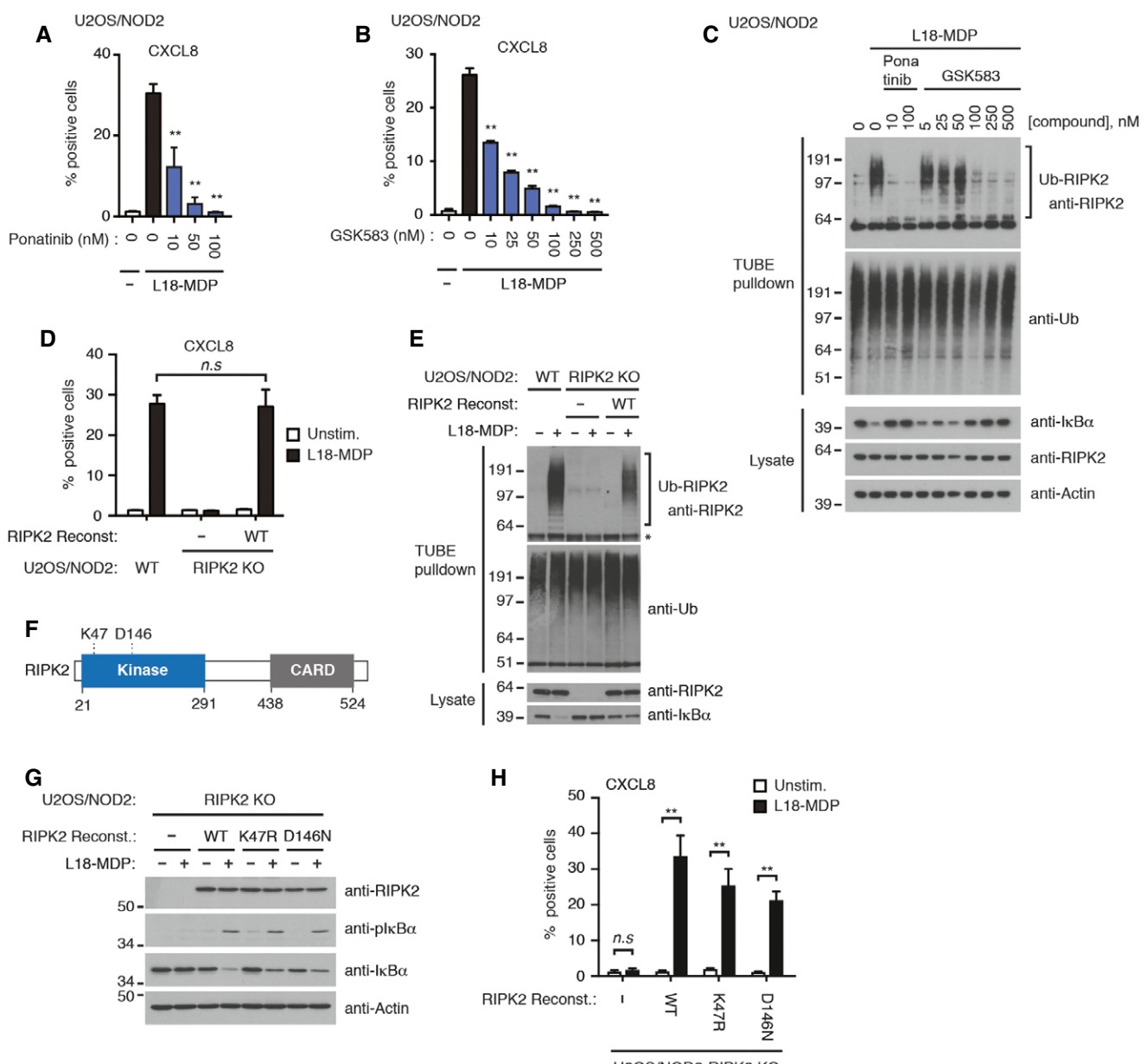

**Figure 1.  RIPK2 kinase activity is dispensable for NOD2 signaling.**

A, B    Intracellular flow cytometry analysis of CXCL8 in U2OS/NOD2 cells treated with L18-MDP (200 ng/ml, 4 h) and kinase inhibitors ponatinib (A) or GSK583 (B) as indicated.

C      Purification of Ub-conjugates using TUBE pulldowns from U2OS/NOD2 cells after treatment with L18-MDP (200 ng/ml, 1 h) and ponatinib or GSK583 as indicated. Purified material and lysates were analyzed by immunoblotting, with actin as a loading control.

D      Intracellular flow cytometry analysis of CXCL8 following L18-MDP treatment (200 ng/ml, 4 h) of parental U2OS/NOD2 cells and RIPK2 KO cells (clone C5-2) reconstituted or not with RIPK2.

E      Purification of Ub-conjugates from parental U2OS/NOD2 cells and RIPK2 KO cells (clone B7-1) reconstituted or not with RIPK2, treated with L18-MDP (200 ng/ml, 1 h). Purified material and lysates were analyzed by immunoblotting. Asterisk indicates a non-specific signal in the TUBE pulldown samples that co-migrates with the signal for unmodified RIPK2.

F      Schematic representation of RIPK2. Numbering below schematic refers to amino acid residues in human RIPK2 and indicate domain boundaries. K47 and D146 are catalytic residues for ATP hydrolysis.

G      Immunoblot analysis of U2OS/NOD2 RIPK2 KO cells (clone C5-2) reconstituted with RIPK2 variants or empty vector as indicated and stimulated (or not) with L18-MDP (200 ng/ml, 1 h).

H      Intracellular flow cytometry analysis of CXCL8 following L18-MDP treatment (200 ng/ml, 4 h) of cells described in (G).

Data information: Data in (A, B, D, H) represent the mean ± SEM of 2–4 independent experiments, each performed in duplicate. Statistical significance in (A) and (B) is determined in relation to L18-MDP-stimulated samples without inhibitor. **$P < 0.01$, n.s., not significant. Two-way ANOVA was used to determine statistical significance. See also Fig EV1.

Source data are available online for this figure.

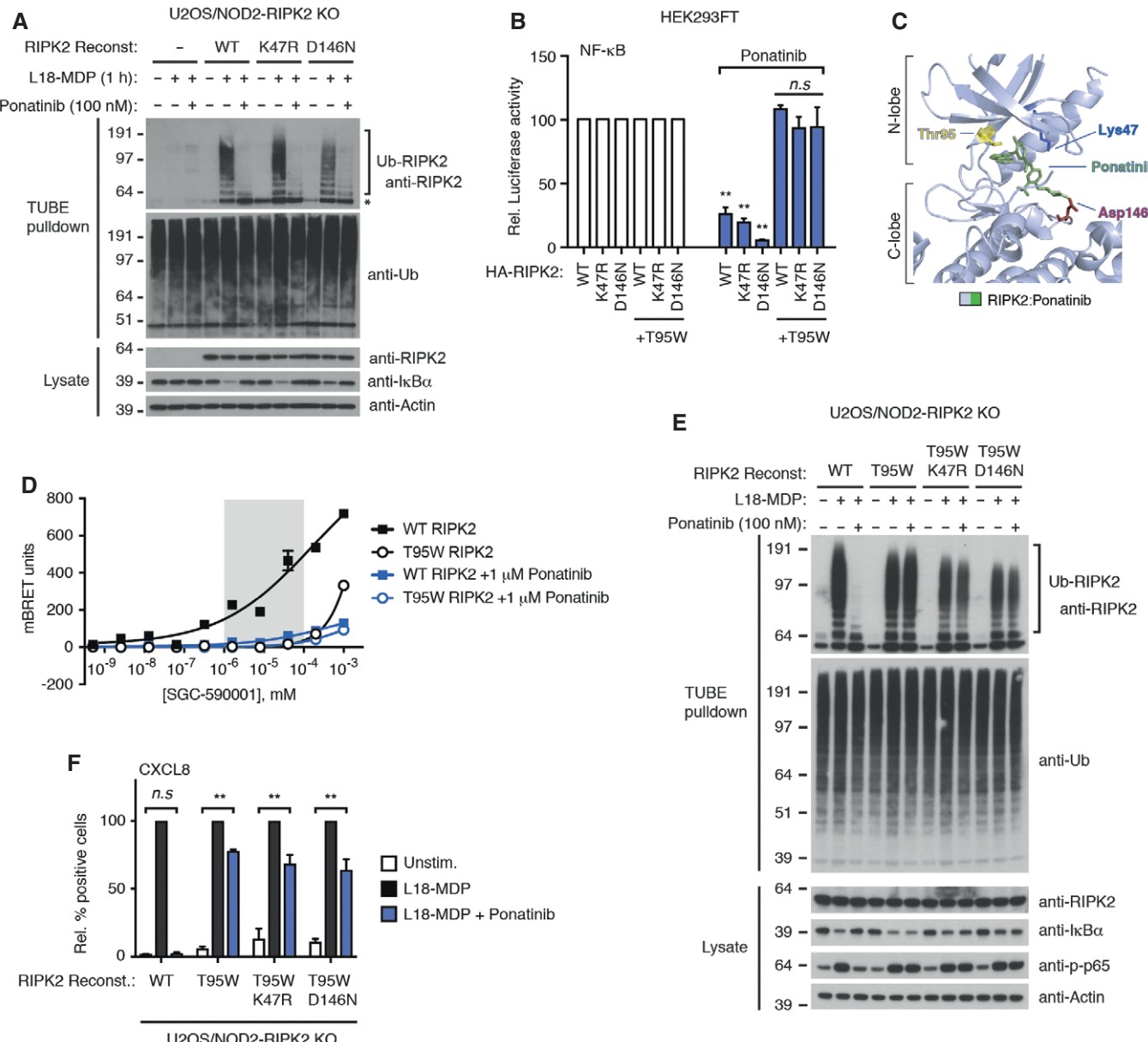

**Figure 2. Ponatinib antagonizes NOD2 signaling through binding to RIPK2 but independently of its kinase activity.**

A Purification of Ub-conjugates from U2OS/NOD2 RIPK2 KO cells (clone C5-2) reconstituted with RIPK2 variants or vector as indicated and treated with L18-MDP (200 ng/ml, 1 h) and/or ponatinib. Purified material and lysates were analyzed by immunoblotting. Asterisk indicates a non-specific signal in the TUBE pulldown samples that co-migrates with the signal for unmodified RIPK2.

B NF-κB activity in lysates of HEK293FT cells transfected with dual luciferase NF-κB reporters and HA-RIPK2, and treated with DMSO or ponatinib (200 nM, 24 h) as indicated. Relative luciferase activity in ponatinib-treated samples is shown relative to the activity in the corresponding HA-RIPK2 transfected sample not treated with inhibitor.

C Structure of the RIPK2 kinase domain in complex with ponatinib (green) (PDB ID: 4C8B). Sticks are shown for catalytic residues Lys47 (blue) and Asp146 (red), and the gatekeeper residue Thr95 (yellow).

D NanoBRET assay in HEK293 cells transiently transfected with the NanoLuc-RIPK2. Cells were treated with serial dilutions of SGC-590001 probe and incubated with ponatinib (1 μM) or DMSO as a control for 3 h before measurement of BRET ratios. Gray box indicates range of ponatinib concentrations used in signaling experiments in this study.

E Purification of Ub-conjugates from U2OS/NOD2 RIPK2 KO cells reconstituted with RIPK2 variants or vector as indicated and treated with L18-MDP (200 ng/ml, 1 h) and/or ponatinib (100 nM). Purified material and lysates were analyzed by immunoblotting.

F Intracellular flow cytometry analysis of CXCL8 of U2OS/NOD2 RIPK2 KO cells reconstituted retrovirally with RIPK2 variants or vector as indicated and treated with L18-MDP (200 ng/ml, 4 h) and ponatinib (50 nM). Values represent CXCL8-positive cells relative to L18-MDP treatment for each RIPK2 variant or empty vector.

Data information: Data in (B, D, F) represent the mean ± SEM of 3–4 independent experiments, each performed in duplicate. **$P < 0.01$, n.s., not significant. Two-way ANOVA was used to determine statistical significance. See also Appendix Fig S1.

Source data are available online for this figure.

CXCL8 (Fig 2E and F). Together, these observations show that ponatinib inhibits NOD2 signaling through its binding to RIPK2 but not by inhibiting the kinase activity of RIPK2.

## Development of a new series of potent small molecule RIPK2 inhibitors

Ponatinib displays highly promiscuous inhibitory activity (Fauster *et al*, 2015; Najjar *et al*, 2015), and clinical development of GSK583 was halted (Haile *et al*, 2016), raising a need for new classes of RIPK2 inhibitors. We have developed a new chemical series of RIPK2 inhibitors, termed CSLP. A subset of these inhibitors, exemplified by CSLP37 and CSLP43, displayed excellent potency in the NOD2/HEKBlue reporter assay, measuring NF-κB activation in response to L18-MDP [Table 1; full details of the synthesis and structure–activity relationship (SAR) will be reported separately (CS, BD, DMP, ALD, LL, MH, LS, MGH, MHU, ANB, AD, GDC, manuscript in preparation)]. However, correlation analysis of an entire panel of CSLP analogs revealed a startling disparity between inhibition of RIPK2 kinase activity *in vitro* and suppression of the NOD2/RIPK2 pathway in cells. Specifically, while many CSLP inhibitors displayed comparably potent activity against RIPK2 kinase activity *in vitro*, only a small subset of these compounds, i.e., CSLP37/43, provided potent suppression of cellular NOD2/RIPK-dependent responses as determined by L18-MDP-induced CXCL8 production in U2OS/NOD2 cells

and NF-κB activation in NOD2/HEKBlue cells (Figs 3A and B, and EV2A; Table 1). Of note, CSLP37/43 and other CSLP inhibitors did not have measurable toxicity in the cell lines used in this study (Fig EV2B). Additionally, CSLP37 and CSLP43 showed no inhibitory activity against the two closest mammalian homologs of RIPK2, RIPK1, and RIPK3 (Fig EV2C).

To identify the reasons for the outstanding cellular activities displayed by CSLP37/43, we focused on a small subset of highly structurally similar CSLP analogs, differing in $R^1$, $R^2$, and $R^3$ substituents, that displayed similar inhibition of RIPK2 *in vitro* but widely variable cellular activity in the NOD2/HEKBlue reporter assay (Table 1). We first examined whether these molecules displayed major differences in binding to RIPK2 in cells by using the nanoBRET RIPK2 target engagement assay described above (Fig 2D). Indeed, while CSLP37/43 displayed a potent target engagement in-line with their activities in the HEKBlue reporter assay, other CSLP inhibitors, such as CSLP18, differing from CSLP37/43 only in the $R^1$ group, CSLP38 (different $R^2$), CSLP55 (different $R^3$), CSLP48 (different $R^1$ and $R^2$), displayed lower target occupancy, correlating with reduced cellular activities (Table 1). These data suggested that the identity of the $R^1$-$R^3$ groups plays a major role in inhibitor binding to cellular RIPK2, which dictates the ability of CSLP inhibitors to suppress NOD1/2 signaling. We also examined target residence time by determining the time required for a nanoBRET probe to engage RIPK2 after washout of the inhibitor from the cells (t1/2) to further

**Table 1.   *In vitro* and cellular activities of CSLP analogs.**

**CSLP**          **WEHI-345**

| Compound ID | X | $R^1$ | $R^2$ | $R^3$ | IC$_{50}$ (nM) | | | |
|---|---|---|---|---|---|---|---|---|
| | | | | | In vitro kinase | | Cellular activity | |
| | | | | | RIPK2 ADPGlo | HEKBlue NOD2 | nanoBRET RIPK2 binding | nanoBRET residence time, min |
| CSLP43 | NH$_2$ | OMe | OMe | -NHSO$_2{}^n$Pr | 19.9 ± 0.8 | 1.3 ± 0.4 | 10.1 ± 3.8 | 106.9 |
| CSLP37 | NH$_2$ | F | OMe | -NHSO$_2{}^n$Pr | 16.3 ± 4.6 | 26.3 ± 3.7 | 36.3 ± 20.2 | 27.1 |
| CSLP18 | NH$_2$ | H | OMe | -NHSO$_2{}^n$Pr | 31.6 ± 8.7 | 476.0 ± 96.7 | 577.6 ± 34.1 | 66.6 |
| CSLP38 | NH$_2$ | F | H | -NHSO$_2{}^n$Pr | 39.1 ± 1.5 | 740.3 ± 60.7 | 166.8 ± 19.0 | 106.5 |
| CSLP48 | NH$_2$ | H | OH | -NHSO$_2{}^n$Pr | 53.5 ± 5.7 | > 5,000 | 1,231.3 ± 344.5 | 126.7 |
| CSLP53 | Me | F | OMe | -NHSO$_2{}^n$Pr | 1,414.5 ± 311.6 | 2,556.5 ± 252.8 | > 10,000 | ND |
| CSLP55 | NH$_2$ | F | OMe | OMe | 39.1 ± 3.9 | 595.1 ± 69.6 | 194.6 ± 47.1 | 6.8 |
| WEHI-345 | | | | | 37.3 ± 1.3 | 3,370.7 ± 382.8 | 521.2 ± 171.6 | 7.5 |

Compounds were tested against recombinant RIPK2 kinase using ADPGlo assay, L18-MDP-induced NF-κB reporter assay (HEKBlue assay), nanoBRET cellular RIPK2 target engagement assay (in HEKBlue cells), and nanoBRET cellular RIPK2 residence time assay (in HEKBLue cells). ND—not determined due to very poor binding of CSLP53 to RIPK2 in nanoBRET assay. For each inhibitor, at least three titrations were performed and data were used to calculate average IC$_{50}$ and SD values. Details of each assays are described in the Appendix Supplementary Methods. Chemical structure of the CSLP scaffold and WEHI-345 is shown in above table.

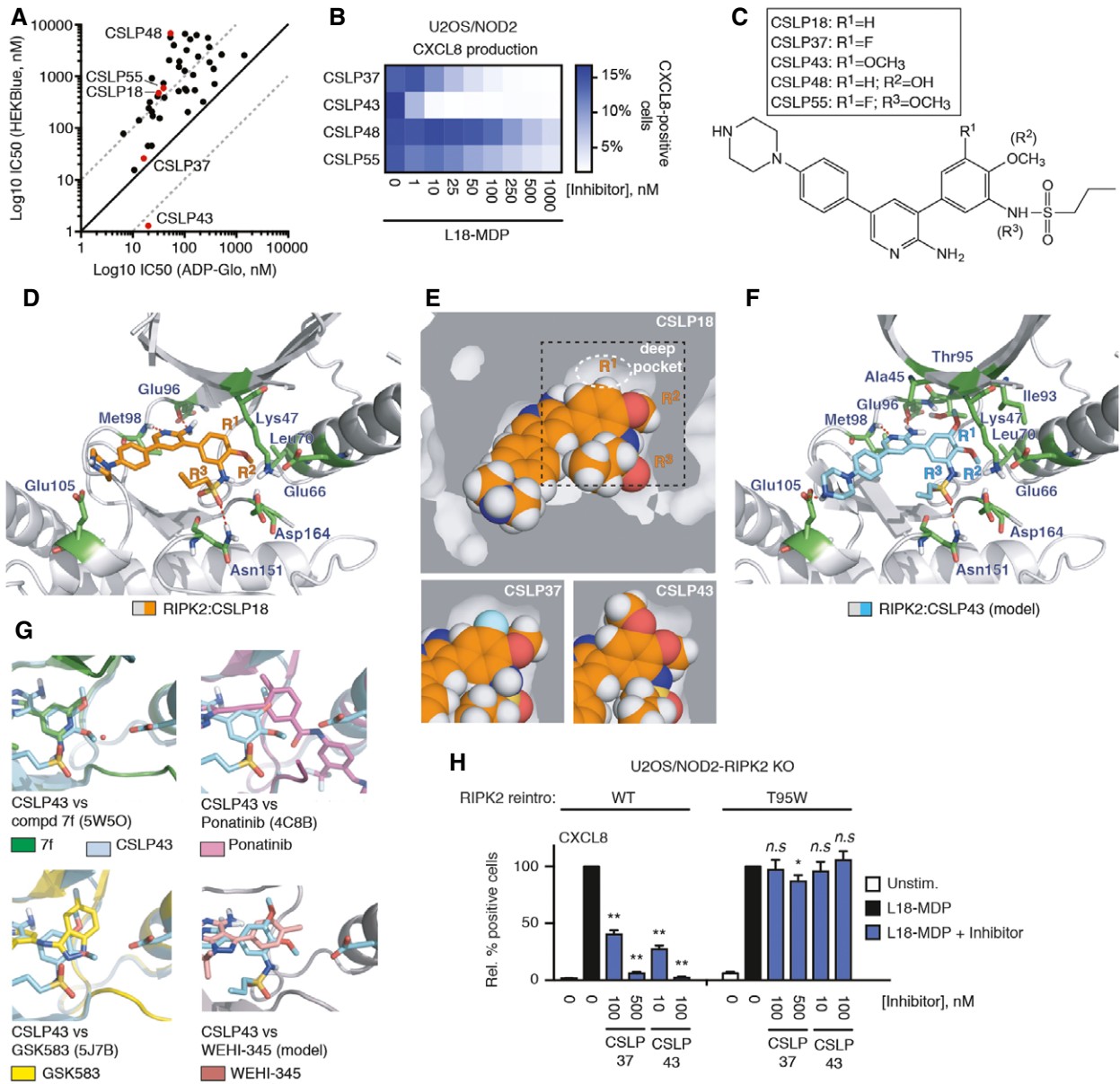

**Figure 3. CSLP series of RIPK2 inhibitors reveal molecular determinants for NOD2 pathway inhibition.**

A   Comparison of inhibitory activity of CSLP compounds on *in vitro* RIPK2 kinase activity (ADPGlo)) and NOD2 signaling in cells (HEKBlue). Compounds characterized further in this study are indicated in red.

B   Intracellular flow cytometry analysis of CXCL8 in U2OS/NOD2 cells treated with L18-MDP (200 ng/ml, 4 h) and CSLP inhibitors as indicated. Data represent the mean of three independent experiments.

C   Chemical structure of CSLP compounds (18, 37, 43) that differ only in R1 group.

D   Structure of RIPK2 kinase domain in complex with CSLP18 (orange) (PDB ID 6FU5). Sticks are shown for catalytic residues Lys47 and Asp146 (in DFG motif), Glu66 forming a salt bridge to Lys47 in active Glu-in conformation, and residues involved in binding of CSLP inhibitors as described in the text.

E   Spacefill rendering of RIPK2 kinase domain structure with CSLP18 (top) and models with CSLP37 (bottom left) and CSLP43 (bottom right). Dark gray represents areas occupied by RIPK2; white areas indicate empty spaces in CSLP binding pocket. Dotted white circle indicates cavity occupied by R1 group of CSLP37/43. Dotted black box indicates region shown for models with CSLP37 and CSLP43.

F   Molecular docking model of RIPK2 kinase domain in complex with CSLP43 (green) based on RIPK2/CSLP18 structure from (D). Key residues from CSLP18/RIPK2 structure and residues forming "R1" pocket, Ala45, Lys47, Ile93, Thr97 are shown as sticks.

G   Comparison of the binding poses of CSLP43 (blue) docking model from (E) based on RIPK2/CSLP18 structure (C) with other RIPK2 kinase inhibitors—Compound 7f (PDB ID 5W5O), ponatinib (PDB ID 4C8B), GSK583 (PDB ID 5J7B), and WEHI-345 (molecular docking model based on RIPK2/CSLP18 structure). While compound 7, ponatinib, GSK583 occupy similar or larger spaces in the deep pocket, WEHI-345 does not contain groups equivalent to R1 and R3 of CSLP43.

H   Intracellular flow cytometry analysis of CXCL8 of U2OS/NOD2 RIPK2 KO cells reconstituted with WT RIPK2 or T95W mutant, and treated with L18-MDP (200 ng/ml, 4 h) and CSLP inhibitors as indicated. Values represent CXCL8-positive cells relative to L18-MDP treatment for each RIPK2 variant without inhibitor treatment.

Data information: Data in (H) represent the mean ± SEM of three independent experiments. *P < 0.05, **P < 0.01, n.s., not significant. Two-way ANOVA was used to determine statistical significance. See also Fig EV2.

elucidate whether the observed differences in potency may reflect changes in off-rates of the inhibitors, but found no correlation (Table 1, nanoBRET residence time). These data suggested that efficient target engagement is a requisite for potent cellular activity of CSLP molecules, which is dictated by the unique combination of $R^1$-$R^3$ substituents.

To further understand the structural contributions of $R^1$-$R^3$, we crystallized RIPK2 in complex with CSLP18 (Table 1), containing a hydrogen (H) in the $R^1$ position compared to fluorine (F) and methoxy (OMe) in CSLP37 and 43, respectively (Fig 3C and D; Table EV1). Overall, the complex with CSLP18 revealed RIPK2 in an inactive conformation previously shown for Type $I_{1/2}$A inhibitors (Wu *et al*, 2015; Roskoski, 2016). Inhibitors in this class interact exclusively within the ATP-binding pocket, in which a number of critical residues are misplaced from their active positions. Specifically, when compared to structures of active RIPK2 in complex with the ATP analog AMP-PNP, the RIPK2-CSLP18 structure showed misorientation of the catalytic residues Asp146 of the HHD motif and Asp164 of the DFG motif, a slight misalignment of the R-spine, a shifted P-loop, and a completely disordered activation segment (AS) which is partially structured in active RIPK2 (Fig EV2D). The structure is refined at 3.2 Å resolution, and the electron density map is of sufficient quality in the region of the inhibitor to place the inhibitor and its relevant functional groups with reasonably good precision (Fig EV2E). CSLP18 forms two hydrogen bonds to the backbone of the hinge segment residues Glu96 and Met98, and the $R^3$ sulfonamide forms a hydrogen bond to Asn151. The $R^3$ hydrogen bond helps position the phenyl ring and $R^2$ OMe group, which inserts into a sub-pocket formed by the side chains of Glu66 and Leu70 of the αC-helix (Fig 3D). The $R^2$ group of CSLP18 fits the binding pocket snugly. However, the $R^1$ position does not optimally fill the hydrophobic pocket (termed deep pocket) formed by the side chains of Ala45, Lys47, Ile93, and gatekeeper Thr95 located on the β3 and β5 strands of the β sheet bundle (Fig 3D and E). This is further revealed by the molecular docking of CSLP37 ($R^1$ = F) and CSLP43 ($R^1$ = OMe), which demonstrates that the larger $R^1$ groups of these molecules more optimally fill the deep pocket (Figs 3E and F, and EV2F). Notably, examining available crystal data, we observed that all previously reported inhibitors that potently antagonize NOD2 signaling, including GSK583, ponatinib, and the recently reported compound 7f (Canning *et al*, 2015; Nachbur *et al*, 2015; Najjar *et al*, 2015; Haile *et al*, 2016; He *et al*, 2017), all occupy the deep pocket (Fig 3G).

Lastly, consistent with this binding mode and similar to our ponatinib data (Fig 2F), inhibition of NOD2 signaling by CSLP37 and CSLP43 was no longer observed in cells expressing a mutant RIPK2 with a bulky tryptophan (W) replacing T95 gatekeeper residue forming part of the pocket occupied by the $R^1$ group (Fig 3F and H). In contrast, inhibition of signaling by CSLP48 and CSLP55, which displayed poor engagement of cellular RIPK2, was not negatively impacted by the T95W mutation (Fig EV2G).

## CSLP37/43 selectively inhibit NOD responses in cells and display potent activity *in vivo*

To further establish cellular activities of CSLP37 and CSLP43, we first confirmed that these molecules efficiently blocked MDP-elicited inflammatory signaling by measuring the release into the media of

TNF (tumor necrosis factor), which is stimulated in macrophages in response to NOD2 activation (Ammann *et al*, 2014). CSLP37 and CSLP43 blocked the release of TNF in these cells with similar $IC_{50}$ values as determined in the NOD2/HEKBlue assay, while CSLP48 and CSLP55 displayed substantially lower inhibitory activities (Figs 4A and B, and EV3A). In contrast, neither CSLP37 nor CSLP43 attenuated the TNF release following an unrelated pro-inflammatory stimulation mediated through Toll-like receptor 4 (TLR4) activation by LPS, even when used at 1 μM (Fig EV3B). Similarly, in experiments with human THP1 monocytes containing an NF-κB-SEAP reporter (THP1Blue), CSLP37 and CSLP43 selectively blocked NOD1 and NOD2 responses at low nanomolar concentrations, with no detected effect on NF-κB responses following stimulation of cells with the non-NOD agonists: TLR1 and TLR2 agonist Pam3CSK4, LPS, or heat-killed Listeria monocytogenes (HKLM), which stimulates mainly TLR2 (Fig 4C).

Nachbur *et al* (2015) described a promising ATP-competitive RIPK2 inhibitor, WEHI-345, which displayed high affinity against RIPK2 kinase *in vitro* but showed > 10-fold lower cellular potency (Table 1; Fig 4A), and only partially inhibited the increase in serum levels of TNF in mice challenged with MDP (Nachbur *et al*, 2015). CSLP37 displayed improved cellular potency over WEHI-345 (Table 1), and when compared side-by-side with WEHI-345 in mice challenged with intraperitoneal injection of MDP, pretreatment with CSLP37 potently reduced MDP-elicited serum TNF levels, whereas WEHI-345 provided a partial reduction (Fig 4D). Overall, these data suggest that CSLP37 and CSLP43 represent a new class of RIPK2 inhibitors, which display comparably high cellular potency to ponatinib and GSK583, strong selectivity toward cellular NOD1/2 responses, and effective inhibitory activity *in vivo*.

## Kinase inhibitors antagonize RIPK2-XIAP interaction to inhibit NOD2 signaling

XIAP and cIAPs ubiquitinate RIPK2 following NOD2 stimulation (Damgaard *et al*, 2012), and the interaction between the XIAP BIR2 domain and the RIPK2 kinase domain is needed for NOD2 signaling (Damgaard *et al*, 2013; Chirieleison *et al*, 2017). Small molecule IAP antagonists such as Compound A (CpA) can block NOD2 signaling by interfering with the interaction between XIAP and RIPK2 and thereby prevent the ubiquitination of RIPK2 by XIAP (Fig 5A; Krieg *et al*, 2009; Damgaard *et al*, 2013; Hrdinka *et al*, 2016). Considering that ponatinib efficiently blocked RIPK2 ubiquitination and that WEHI-345 interferes with the IAP-RIPK2 interaction (Nachbur *et al*, 2015), we hypothesized that ponatinib may inhibit NOD2 signaling by interfering with the XIAP-RIPK2 binding interface. In support of this, pretreatment of cells with ponatinib prevented the co-immunoprecipitation of endogenous RIPK2 with ectopically expressed HA-tagged XIAP (Fig 5A). Furthermore, addition of ponatinib to cell lysates interfered with the ability of recombinant GST-tagged XIAP BIR2 domain (GST-BIR2-XIAP) bound to Glutathione Sepharose to pull down RIPK2 (Figs 5B and EV4A). Strikingly, addition of GSK583, CSLP37, or CSLP43 to cell lysates also efficiently antagonized pulldown of RIPK2 by GST-BIR2-XIAP. whereas CSLP48 and CSLP55, which displayed lower potency in the cells due to the poor occupancy of the deep pocket of RIPK2, had little effect (Fig 5C). The region in the RIPK2 kinase domain (KD) responsible for binding

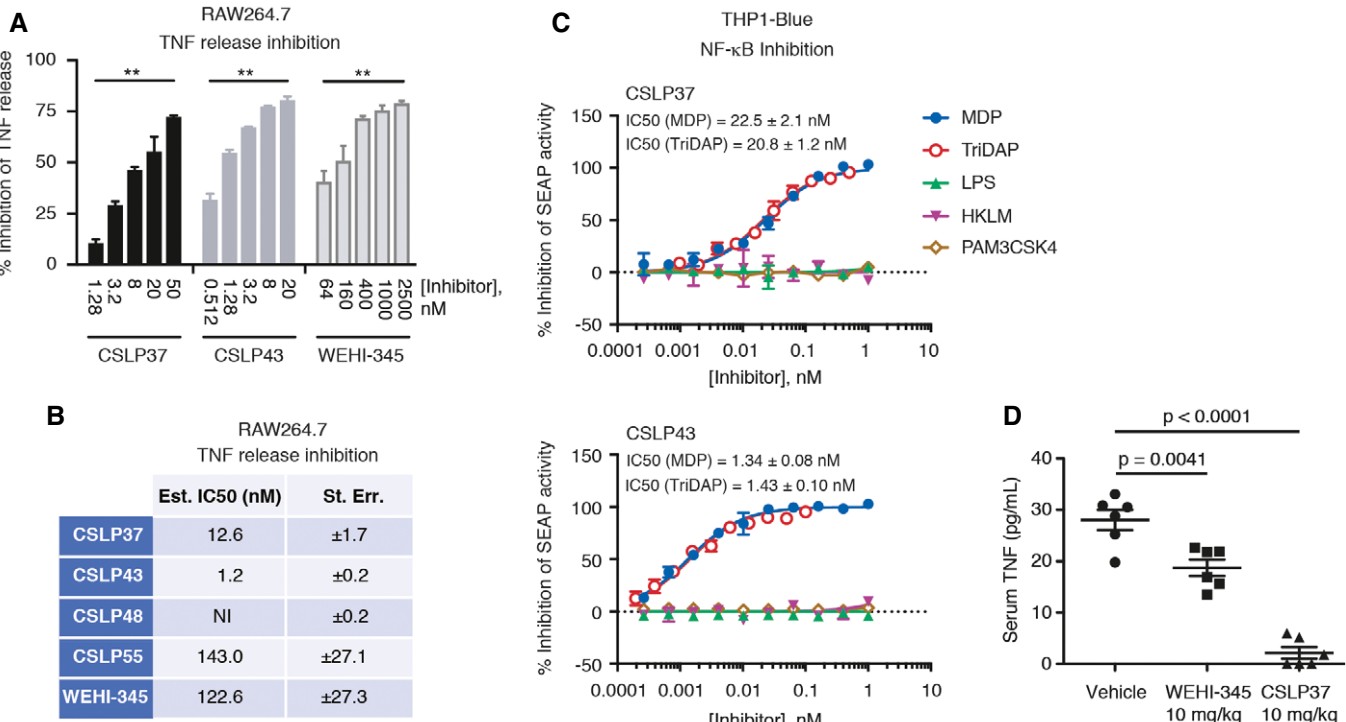

**Figure 4. Cellular and *in vivo* characterization of novel RIPK2 inhibitors on inflammatory signaling.**

A, B   ELISA measurement of TNF release from RAW264.7 cells treated with MDP (10 μg/ml, 24 h) and CSLP compounds or RIPK2 inhibitor WEHI-345 as indicated. Values are expressed relative to the TNF release in cells treated with MDP without inhibitor (A) and are used for calculation of $IC_{50}$ values (B).

C   Inhibition of NF-κB reporter activity by CSLP37 (top) and CSLP43 (bottom) in THP1Blue cells treated with PRR ligands: MDP (10 μg/ml); TriDAP (10 μg/ml); *E. coli* LPS (10 ng/ml); HKLM (1 × 10⁷ cells/ml; or Pam3CSK4 (10 ng/ml). Inhibition values are used for calculating $IC_{50}$ of MDP and TriDAP responses as shown.

D   Inhibition of MDP-elicited TNF release by CSLP37 *in vivo*. Mice (n = 6 per group) were administered i.p. with 10 mg/kg of WEHI-345 or CSLP36 30 min prior to the i.p. injections of 100 μg/mouse of MDP. After 4 h, blood was collected by cardiac puncture and circulating levels of TNF were analyzed by ELISA.

Data information: Data represent the mean ± SEM of three independent experiments. **$P < 0.01$, n.s., not significant. Two-tailed unpaired Student's *t*-test was used to determine statistical significance. See also Fig EV3.

---

to the XIAP BIR2 domain is not known, and the interaction may therefore be either direct or indirect, involving another cellular component. To test this, GST-BIR2-XIAP pulldown experiments were performed with recombinant human RIPK2 kinase domain (KD) instead of cell lysates. Under these conditions, GST-BIR2-XIAP binding to RIPK2 was maintained and was still antagonized by GSK583, CSLP37, and CSLP43, indicating that XIAP and RIPK2 interact directly through the XIAP BIR2 domain and the RIPK2 KD (Fig 5D). cIAP1 also interacts with RIPK2 via its BIR2 domain, and the interaction can be antagonized by WEHI-345 (Nachbur *et al*, 2015). Akin to this, CSLP37 and CSLP43 both antagonized the interaction between GST-BIR2-cIAP1 and RIPK2 in cell lysates (Fig EV4B), suggesting that cIAP1 and XIAP interact with RIPK2 through a similar mechanism. In concordance with the observed potency of CSLPs to interfere with the XIAP-RIPK2 interaction, CSLP37 and CSLP43 inhibited RIPK2 ubiquitination in cells stimulated with L18-MDP, while CSLP48 and CSLP55 again had little or no effect (Fig 5E and EV4C and D). Taken together, these data suggest that potent inhibition of NOD2 cellular responses by ponatinib, GSK583, CSLP37, and CSLP43 involves interference with the RIPK2-IAP binding interface and therefore the critical step of RIPK2 ubiquitination by XIAP.

## Mapping of XIAP/RIPK2 binding interface

Having determined that the BIR2 domain of XIAP interacts directly with the RIPK2 kinase domain, we set out to map the region in RIPK2 responsible for the interaction. For this, we analyzed by SPOT assay the binding of recombinant GST-BIR2-XIAP to 15-mer peptides derived from a walk-through of the entire RIPK2 coding sequence with a frameshift of 3 amino acids (Fig EV5A; Dataset EV2). Seven putative interacting peptides were identified in the kinase domain of RIPK2 (Fig EV5A, blue squares; Fig EV5B). These seven peptides were printed in triplicate on a new array and probed with either GST-tagged WT XIAP BIR2 domain or a BIR2 domain containing a mutation in the IAP-binding motif groove (D214S) that interferes with RIPK2 binding (Figs 6A and EV4A, Dataset EV3; Damgaard *et al*, 2013). This showed that the WT BIR2 domain binds predominantly to two RIPK2 peptides, A13 (residues 28-42), and E6 (residues 163–177; Fig 6A and B). Comparison of the binding of WT versus D214S BIR2 domain showed that only the interaction with peptide A13 (position B1, D1, and F1) was reduced when probed with the D214S mutant (Fig 6A and B). This region spans β2 and the following β2-β3 loop of the N-lobe of the RIPK2 kinase domain (Fig 6C and D). Strikingly, single-residue Ala- and Leu-scanning

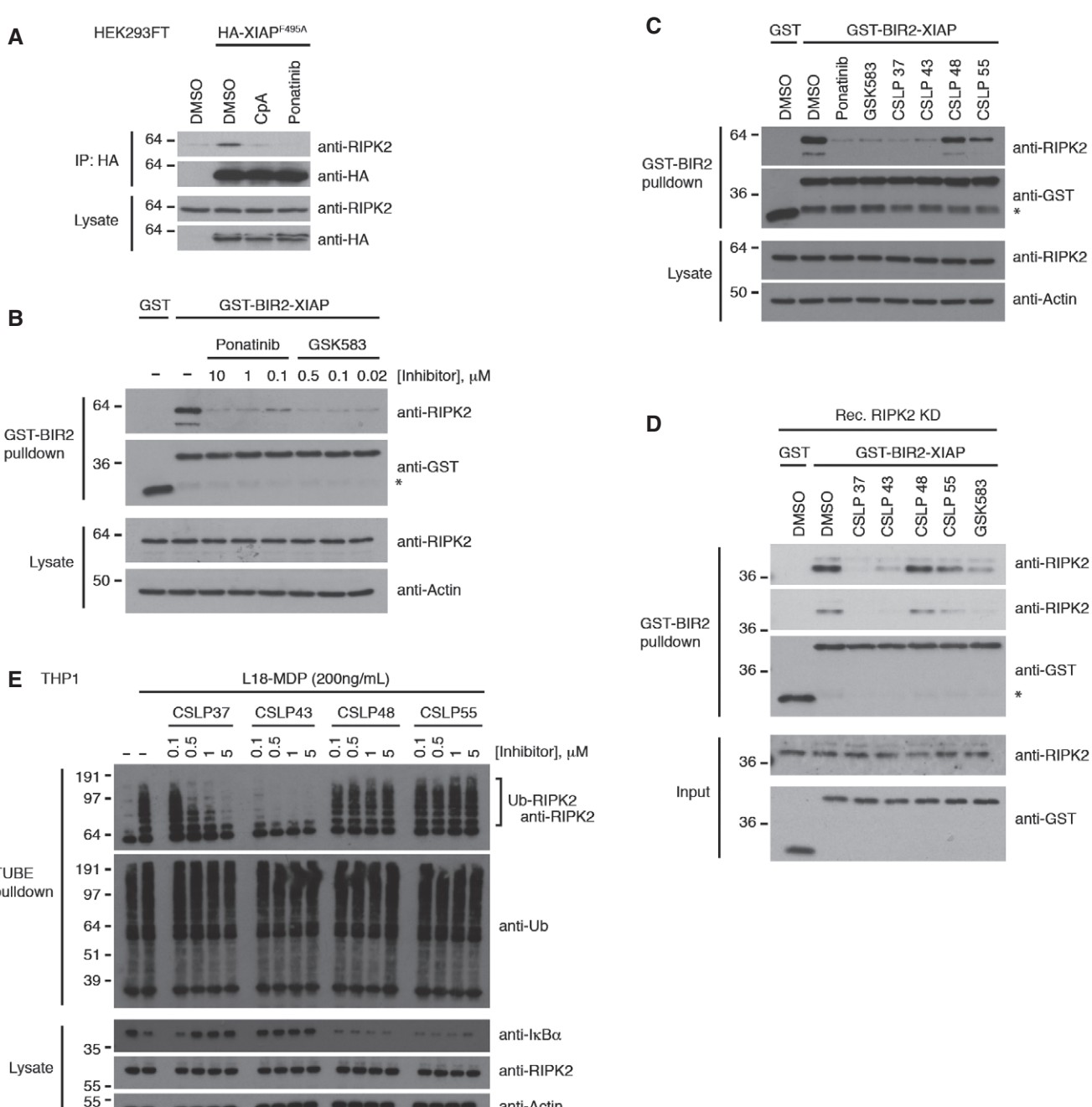

**Figure 5.   RIPK2 inhibitors antagonize NOD2 signaling by interfering with the XIAP-RIPK2 interaction.**

A   Co-immunoprecipitation and Western blot analysis of HA-XIAP (F495A; Ub ligase-dead) and endogenous RIPK2 from HEK293FT cells treated with IAP antagonist compound A (CpA, 10 μM) or ponatinib (200 nM).

B   Pulldown of RIPK2 from U2OS/NOD2 cell lysates with recombinant GST-tagged BIR2 domain (residues aa 124–242) from XIAP (GST-BIR2-XIAP) in the presence of ponatinib or GSK583 as indicated. Purified material and lysates were analyzed by immunoblotting.

C   Pulldown of RIPK2 from U2OS/NOD2 cell lysates with recombinant GST-BIR2-XIAP in the presence of CSLP inhibitors. Inhibitors were used at 100-fold $IC_{50}$ of RIPK2 kinase activity: ponatinib (10 μM), GSK583 (5 μM), CSLP37 (1.8 μM), 43 (2 μM), 48 (5.7 μM), and 55 (3.9 μM) as indicated. Purified material and lysates were analyzed by immunoblotting.

D   Pulldown of recombinant RIPK2 kinase domain (dephosphorylated) with recombinant GST-BIR2-XIAP in the presence of CSLP inhibitors as indicated. Inhibitors were used at the same concentrations as in (C). Purified material and lysates were analyzed by immunoblotting.

E   Purification of Ub-conjugates from THP-1 cells after treatment with L18-MDP (200 ng/ml, 1 h) and CSLP inhibitors as indicated. Purified material and lysates were analyzed by immunoblotting.

Data information: Asterisk in (B–D) indicates a cleavage product of GST-BIR-XIAP recognized by the GST antibody. See also Fig EV4.

Source data are available online for this figure.

analysis of the A13 peptide and an overlapping peptide covering the β2-β3 loop and β3 (residues 37-51) revealed that mutation of R36, H37, or R41 impaired BIR2 binding (Figs 6E and EV5C, Dataset EV4). To investigate the function of the R36/R41 patch, we reconstituted U2OS/NOD2 RIPK2 KO cells with RIPK2 mutants in which R36 and/or R41 were substituted with leucine (Fig EV5D). Mutation of these residues impaired the interaction with GST-BIR2-XIAP in pulldown experiments with lysates from the reconstituted cells (Fig 6F). Accordingly, the RIPK2 R41L single and R36L/R41L double mutants failed to fully restore NOD2 signaling in the reconstituted cells (Fig 6G). Intriguingly, R36 and R41 are surface-exposed and form a basic patch positioned on top of the deep pocket occupied by the compounds that potently inhibit NOD2 signaling (Figs 6H and EV5E), implying that the compounds alter the ability of XIAP to interact with the R36/R41 patch (Fig 6D).

## Discussion

Targeting NOD1 and NOD2 signaling with RIPK2 inhibitors is an attractive strategy for the treatment of chronic inflammatory conditions associated with deregulated NOD signaling. These conditions include hereditary auto-inflammatory syndromes associated with activating mutations in NOD2, such as early-onset sarcoidosis and Blau syndrome, and early-onset inflammatory bowel disease, in particular Crohn's disease where NOD2 signaling may play an important role (Kanazawa *et al*, 2005; Stronati *et al*, 2008; Negroni *et al*, 2009; Yao *et al*, 2011; Caruso *et al*, 2014; Uhlig & Schwerd, 2016). XIAP-mediated ubiquitination of RIPK2 emerged as a requisite step in pro-inflammatory signaling downstream of NOD1 and NOD2 (Bauler *et al*, 2008; Damgaard *et al*, 2012), and XIAP-selective IAP antagonists have been recently reported to efficiently impair NOD signaling (Goncharov *et al*, 2018). However, continuous inhibition of XIAP as a therapeutic strategy raises concerns due to the role of this protein in innate immune and cell death regulation and because the loss-of-function mutations of XIAP associate with hereditary very early-onset (VEO) IBD and X-linked

lymphoproliferative disease type 2 (XLP-2) (Rigaud *et al*, 2006; Damgaard *et al*, 2013; Speckmann *et al*, 2013; Latour & Aguilar, 2015).

Initial notions that RIPK2 kinase activity facilitates NOD1 and NOD2 signaling were in part based on the finding that kinase activity may be required for RIPK2 stability in cells, and in part, on the fact that kinase inhibitors could inhibit productive NOD1 and NOD2 signaling (Windheim *et al*, 2007; Nembrini *et al*, 2009; Tigno-Aranjuez *et al*, 2010). However, while several promiscuous clinical Type II inhibitors, such as ponatinib, regorafenib, and sorafenib (Canning *et al*, 2015) displayed very potent (low nanomolar) cellular activity against NOD1- and NOD2-mediated responses, a number of Type I ATP-competitive inhibitors, such as gefitinib and WEHI-345, showed reduced potency in cellular assays despite being comparably potent to Type II molecules in inhibiting RIPK2 activity *in vitro* (Tigno-Aranjuez *et al*, 2010; Canning *et al*, 2015; Nachbur *et al*, 2015). In light of these discrepancies, in the present work we sought to define the features of inhibitors necessary for cellular potency against NOD responses.

We started by evaluating the role of RIPK2 catalytic activity in NOD2 signaling. Strikingly, using reconstitution with two different kinase-dead mutants of RIPK2 we discovered that RIPK2 kinase activity is dispensable for NOD2 inflammatory signaling. Furthermore, ponatinib was still able to completely prevent these responses even in the cells lacking catalytic activity of RIPK2, clearly indicating that efficient inhibition of NOD2 responses is not related to the ability of the inhibitors to block the catalytic activity of RIPK2. Notably, Goncharov *et al* (2018) recently undertook similar experiments and reached the same conclusion that catalytic activity of RIPK2 is not required for NOD2 responses.

As an alternative, we explored whether selected inhibitors may interfere with RIPK2 ubiquitination by XIAP, which is indispensable for NOD1 and NOD2 signaling (Krieg *et al*, 2009; Damgaard *et al*, 2012, 2013). Indeed, we show that all potent NOD pathway inhibitors, including ponatinib, GSK583, and the newly developed CSLP37 and CSLP43 efficiently abrogate both XIAP binding to RIPK2 and RIPK2 ubiquitination. Furthermore, using a panel of closely related

▶

**Figure 6. Mapping and functional characterization of XIAP-binding site on RIPK2.**

A, B   SPOT peptide array of seven putative interacting peptides of the RIPK2 kinase domain, identified in Fig EV5A, probed with purified GST-XIAP-BIR2 (WT or D214S; RIPK2 interaction-defective mutant), and visualized by anti-His-HRP (A). The peptides were spotted and analyzed in triplicates (blue boxes). 11xHis was spotted in triplicate as a positive control for anti-His-HRP staining (yellow boxes). Table (right) indicates the position of each peptide on the membrane and the amino acids in human RIPK2 covered by each peptide. Heat map presentation of each spot in the peptide array quantified using ImageJ (B). Values represent the relative intensity of the mean of each peptide triplicate to the mean of the positive control triplicate in the peptide array.

C       Primary sequence alignment of the N-terminal region in RIPK2, including the region represented in A13 peptide. Alignment shows high evolutionary conservation of the region. Residues R36 and R41 studied for XIAP binding are shown in blue.

D       Cartoon representation of the structure of RIPK2 kinase domain (PDB ID: 5AR2). Region represented in A13 peptide is shown in blue. Sticks are shown for R36 and R41 to highlight their surface orientation. RIPK2 inhibitor GSK583 (yellow) is modeled into structure based on PDB ID: 5J7B (Haile *et al*, 2016).

E       SPOT peptide array of the A13 peptide in which each residue is substituted with alanine (Ala-scan) or leucine (Leu-scan) was probed for binding of GST-XIAP-BIR2. Each peptide is spotted in triplicate indicated by #1-#3. Yellow boxes indicate the residues in A13 where binding is reduced by modification. The bound BIR2 domain was visualized by anti-His-HRP.

F       Pulldown of RIPK2 variants from U2OS/NOD2 cell lysates with recombinant GST-BIR2-XIAP. Purified material and lysates were analyzed by immunoblotting.

G       Intracellular flow cytometry analysis of CXCL8 in U2OS/NOD2 RIPK2 KO cells reconstituted with RIPK2 variants following stimulation (or not) with L18-MDP (200 ng/ml, 4 h). Values represent fraction of CXCL8-positive cells. Data represent the mean ± SEM of five independent experiments.

H       Structure of XIAP-BIR2 (PDB ID: 1C9Q) and RIPK2 kinase domain (KD) (PDB ID: 5AR2). RIPK2 inhibitor GSK583 (yellow) is modeled into the structure based on (PDB ID: 5J7B) (Haile *et al*, 2016). Surface charge representation shows an acidic patch around residues E211, D214, and E219 in XIAP and a basic patch generated by R36 and R41 in RIPK2 (black dotted circles). IAP-binding motif (IBM) groove is indicated by green dotted line in cartoon representation. The inhibitor-binding pocket is indicated by red dotted line.

Data information: **$P < 0.01$, n.s., not significant. Two-way ANOVA was used to determine statistical significance. See also Fig EV5.
Source data are available online for this figure.

CSLP analogs, we show that inhibition of RIPK2 ubiquitination does not generally reflect the affinity of these molecules toward the RIPK2 kinase domain *in vitro*.

This leaves open a critical question of what defines the ability of some, but not all RIPK2 inhibitors to block cellular signaling by interfering with binding of the XIAP BIR2 domain to RIPK2. The crystal structure of one of the CSLP analogs, CSLP18, with RIPK2 provided key insights into this. The RIPK2-CSLP18 complex

structure revealed the inhibitor occupying the ATP pocket of RIPK2 with the $R^1$-$R^3$ groups of the inhibitor inserted into the deep pocket behind the adenine-binding site. Structure–activity relationship analysis of a subset of closely related CSLP analogs revealed that the $R^1$-$R^3$ groups dictate high cellular activity of CSLP analogs (Fig 3F). Molecular docking analysis showed that two methoxy groups in the $R^1$ and $R^2$ positions provide an optimal fit in the deep pocket of RIPK2, leading to low nanomolar cellular activity of CSLP43.

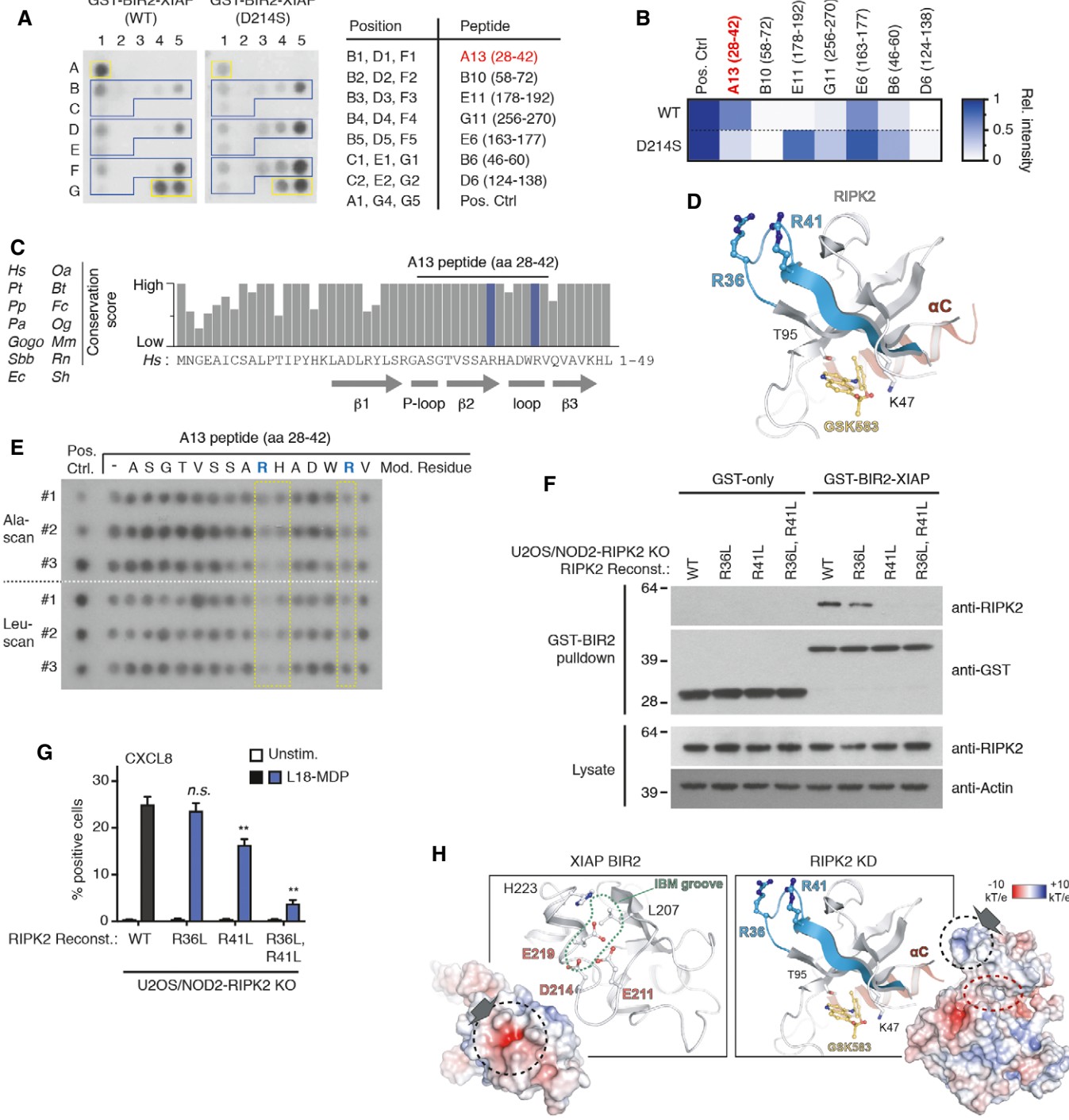

**Figure 6.**

Consistently, molecules lacking $R^1$ or $R^2$, or possessing smaller (CSLP37, 18, 38, 48; Table 1) or slightly larger (such as $R^1$ = Cl or $R^2$ = OEt, data not shown, CS, BD, DMP, ALD, LL, MH, LS, MGH, MHU, ANB, AD, GDC, manuscript in preparation) $R^1/R^2$ groups display greatly reduced cellular activity. Additionally, molecules lacking the hydrogen bond acceptor sulfonamide in $R^3$ (CSLP55; Table 1) also loose cellular activity. The binding of CSLP analogs to RIPK2 in intact HEKBlue cells, measured by nanoBRET assay, closely correlated with the functional activity of the inhibitors in blocking NOD2 signaling. In sum, these data suggested that occupancy of the deep pocket is essential for the ability of CSLP analogs to bind endogenous RIPK2 and to block signaling by interfering with XIAP binding. This "cellular-specific" role of the deep pocket of RIPK2 contrasts with the more conventional changes that affect affinity of the inhibitors toward RIPK2, such as the absence of the hinge-interacting amino group in CSLP53, leading to the loss of both *in vitro* and cellular activities.

This mode of activity may also be relevant for other inhibitors since an alignment of published RIPK2 structures indicates that all potent inhibitors of NOD signaling such as ponatinib (Canning *et al*, 2015) and GSK583 (Haile *et al*, 2016) extend into the deep pocket occupied by $R^1$-$R^3$ substituents of the CSLP series (Fig 3G). Another recently described ATP-competitive RIPK2 inhibitor series with potent activity in cells (e.g., compound 7f) possesses a phenyl ring similar to the CSLP inhibitors (He *et al*, 2017). While this molecule lacks a group corresponding to the $R^2$ group in the CSLP series, it contains a pyridine and the crystal structure indicates the presence of a water molecule in place of $R^2$ in this case (Fig 3G). Conversely, the previously reported Type I inhibitor WEHI-345 likely does not fully occupy the deep pocket as it lacks both $R^1$ and $R^3$ (Fig 3G), which possibly explains its reduced cellular activity (Nachbur *et al*, 2015). Overall, these results reveal that while it is possible to inhibit NOD responses using molecules targeting the ATP-binding pocket, engaging the deep pocket of RIPK2 is critical for the potency of the molecules.

In a complementary set of data, we reveal the molecular interface between RIPK2 and XIAP. Binding data and functional assays suggest that the basic patch formed by R36/R41 located in the N-terminal loop connecting β2 and β3 mediates XIAP binding via a direct, electrostatic interaction with the acidic patch formed by E211, D214, E219 in the BIR2 IBM groove (Fig 6I). This discovery has multiple significant implications. First, it explains why SMAC and some IAP antagonists disrupt the XIAP-RIPK2 interaction and inhibit NOD2 signaling since these molecules dock into the pocket in the IBM groove and thereby occlude the acidic patch (Krieg *et al*, 2009; Damgaard *et al*, 2013; Figs 6H and EV5F). It also explains why charge-changing mutation of the key IBM groove residues D214 (D214S) and E219 (E219R) interfere with RIPK2 binding, and, possibly, why mutation of the basic H223 residue to a hydrophobic residue (H223V) enhances RIPK2 binding (Damgaard *et al*, 2013). Second, identification of the RIPK2/XIAP binding interface offers an opportunity for developing new generations of inhibitors that may possess selectivity toward this interaction as opposed to general inhibition of XIAP functions, some of which may be critical. Third, it raises the question whether the deep pocket of RIPK2 (formed by β3/β5) may possess additional importance because it is located in close proximity and on the opposite side of β3 from the XIAP binding site (Fig 6H). One can envision that if XIAP binding is associated with movement of β3, molecules occluding the deep pocket in an

allosteric fashion could interfere with the binding. Contrary to this, RIPK2's ability to mediate NOD2 signaling does not appear to be sensitive to conformational changes elsewhere in the ATP-binding pocket. Pellegrini *et al* (2017) recently demonstrated significant conformational changes in the ATP-binding pocket associated with the inactivating mutations D146N and K47R and, yet, we find that both mutants retain the ability to be ubiquitinated by XIAP and mediate signaling in response to NOD2 stimulation. In accordance with this, structural comparison of the RIPK2 R36/R41 XIAP-binding area indicates there are no significant changes in the conformation between WT RIPK2 and the mutant proteins (Fig EV5E).

IAP family members have emerged as important mediators of innate immune receptor responses by ubiquitinating RIPKs within signaling complexes. RIPK2 binds to and is ubiquitinated by both XIAP and cIAP1, but only the interaction with XIAP is required for NOD1 and NOD2 inflammatory signaling, whereas cIAPs are dispensable (Damgaard *et al*, 2013; Stafford *et al*, 2018). Nonetheless, recent work suggests that TNF-induced ubiquitination of RIPK1 by cIAPs acts in concert with XIAP-mediated RIPK2 ubiquitination *in vivo* to provide a full innate immune response to bacterial peptidoglycans (Stafford *et al*, 2018). Interestingly, small molecule RIPK2 inhibitors interfere with cIAP1-binding comparably to XIAP, suggesting that these ubiquitin ligases likely share the binding interface on RIPK2. It is therefore plausible that cIAPs contribute to as yet to be defined RIPK2-mediated processes. Other innate immune kinases, such as IRAK1 (Interleukin-1 (IL-1) receptor-associated kinase 1) and IRAK4, are ubiquitination targets for other Ub ligases such as TRAF6 and Pellino1 in TLR/IL-1R signaling (Cohen & Strickson, 2017). Remarkably, knowledge about the substrates of RIPK and IRAK kinases is limited and there is a continuing debate about the role of these proteins as true kinases versus their scaffolding function, including as scaffolds for ubiquitination (Kawagoe *et al*, 2007; Kim *et al*, 2007; Koziczak-Holbro *et al*, 2008; Ordureau *et al*, 2008; Song *et al*, 2009; Vollmer *et al*, 2017; Wang *et al*, 2017; Annibaldi & Meier, 2018; Dziedzic *et al*, 2018). Our findings for RIPK2 exemplify the cross-talk between the kinase domain serving as a binding interface for a Ub ligase (XIAP) and kinase inhibitors acting as blockers of this interaction. This resolves a paradox of how ATP-competitive inhibitors of kinase activity may serve to block a scaffolding catalysis-independent function of the protein. This example may be applicable to other kinases, including other innate immune kinases, whose role in signaling remains enigmatic.

## Materials and Methods

Please see Appendix Supplementary Methods for further details on all methods.

### Generation of knockout and reconstituted cell lines

U2OS/NOD2 RIPK2 KO cells were generated using CRISPR/Cas9 KO plasmids (containing gRNA, Cas9, and EGFP reporter; Santa Cruz Biotechnology) as described in Elliott *et al* (2016). Reconstitution with RIPK2 variants was performed by retroviral transduction using pBABE-Puro plasmids and the retroviral packaging cell line Phoenix-A. Cells were selected with 1 μg/ml puromycin (Invivo-Gen) for 1 week.

## hNOD2-HEKBlue and THP1Blue assays

For HEKBlue assay, cells were seeded into clear 96-well plates at $7.5 \times 10^3$ cells/well and allowed to attach for 48 h. On the day of the experiment, media was changed to 100 µl of HEKBlue detection media (InvivoGen). Inhibitors were diluted and added in 0.5 µl DMSO 15 min prior to the addition of 1 ng/ml L18-MDP (InvivoGen). After 8–9 h, absorbance at 620 nm was measured using Victor3V plate reader (Perkin Elmer, Waltham, MA). Values of media-only wells were subtracted and %inhibition for each compound concentration relative to the DMSO/L18-MDP-treated controls was calculated. Inhibition values ± SD were fitted by non-linear regression using Prism software (GraphPad Software, La Jolla, CA) to calculate $IC_{50}$ values. For THP1Blue assay, $1 \times 10^5$ cells were seeded in 200 µl of RPMI media in 96-well plates and treated with inhibitors in 0.5 µl DMSO for 15 min prior to the addition of 10 µg/ml MDP, 10 ng/ml Pam3CSK4, 10 ng/ml *E. coli* LPS, or $1 \times 10^7$ cells/ml heat-killed *L. monocytogenes* (InvivoGen) for 24 h. After induction, 20 µl of media was mixed with 180 µl of QUANTI-Blue reagent (InvivoGen) and incubated at 37°C for 3–4 h, followed by absorbance measurement at 620 nm. $IC_{50}$ values were calculated as described for HEKBlue assay.

## Inhibition of MDP-elicited TNF release *in vivo*

Female 6- to 8-week-old C57Bl6/J mice (Jackson labs) were administered with the compounds i.p. 30 min prior to the i.p. injections of MDP as previously described (Nachbur *et al*, 2015). After 4 h, blood was collected by cardiac puncture. 100 µl of serum (diluted twofold) per animal was analyzed using anti-mouse TNF ELISA (Thermo-Fisher Scientific).

## Recombinant protein expression and purification and ADPGlo kinase assays

The BIR2 domains of XIAP and cIAP1 were cloned into the pGEX-6P1 vector and were expressed in *E. coli* (BL21) cells as a fusion protein with N-terminal 6xHis (XIAP BIR2 only) and GST tags. Protein expression was induced with IPTG, and cells were lysed using BugBuster protein extraction reagent (#70921, Merck Millipore) according to manufacturer's instructions. Proteins were purified with the FPLC protein purification system ÄKTA PrimePlus (GE Healthcare Life Sciences) using HisTrap FF column purification followed by gel filtration (HiLoad 26/600 Superdex 75 pg (GE Healthcare Life Sciences)). Collected fractions were stored at −80°C and used for probing of peptide array membranes. Expression and purification of recombinant RIPK1-3 was performed as previously described (Canning *et al*, 2015; Najjar *et al*, 2015). RIPK1-3 ADPGlo kinase assays using 20 ng/well protein were performed for 2 h at room temperature also as previously described (Canning *et al*, 2015; Najjar *et al*, 2015).

## Immunoprecipitation, pulldowns, and purification of endogenous ubiquitin conjugates

For co-immunoprecipitation of HA-XIAP with RIPK2, transfected and treated cells were lysed in TBS buffer containing 0.5% NP-40, cOmplete and PhosSTOP inhibitors (Roche) and the lysate was cleared by centrifugation. The lysates were incubated with anti-HA-agarose overnight, washed 3× with lysis buffer, and analyzed by immunoblotting. For GST pulldown experiments, U2OS/NOD2 cells were lysed in TBS lysis buffer containing 0.5% NP-40, cOmplete and PhosSTOP inhibitors (Roche). Cleared lysates were pretreated with kinase inhibitors or DMSO and incubated with GST-XIAP-BIR2 bound to Glutathione Sepharose at 4°C overnight. Bound material was washed 3× with lysis buffer or PBS, eluted with 15 mM Glutathione in PBS, and analyzed by immunoblotting. In pulldown experiments with recombinant (dephosphorylated) RIPK2 kinase domain (Canning *et al*, 2015), inhibitors or DMSO control in PBS were added together with RIPK2 kinase domain to GST-XIAP-BIR2 bound to Glutathione Sepharose. The bound material was eluted and analyzed as described above. Ubiquitin conjugates were purified from treated cells using GST-1xUBA$^{ubq}$ ubiquitin affinity reagent (termed TUBE pulldown) and analyzed by immunoblotting as described (Fiil *et al*, 2013).

## Crystallization and structure determination

Crystals were grown using the vapor-diffusion in sitting drops. The RIPK2:CSLP18 structure was determined as previously described (Canning *et al*, 2015).

## SPOT peptide assays

Cellulose-bound peptide arrays were prepared employing standard Fmoc solid-phase peptide synthesis using a MultiPep-RSi-Spotter (INTAVIS, Köln, Germany) as previously described (Picaud & Filippakopoulos, 2015). The assay was performed using recombinant 6xHIS-GST-BIR2 domain of XIAP protein, and bound protein was detected using anti-His antibody HRP-conjugated. Intensity quantification of SPOTs was performed using ImageJ software (Schneider *et al*, 2012).

## Dual luciferase reporter assay

Cells were co-transfected the NF-κB luciferase reporter construct pBIIX-luc and a thymidine kinase-renilla luciferase construct. Additional plasmids were transfected where indicated and assay performed as described in Damgaard *et al* (2012).

## RIPK2 nanoBRET assay

HEK293 cells were transiently transfected with the NanoLuc-RIPK2 WT and T95W mutant RIPK2 constructs, plated into white 384-well assay plates (Corning, Corning, NY), treated with serial dilutions of SGC-590001, and incubated with 1 µM ponatinib or DMSO control for 3 h. After addition of Nano-Glo Substrate (Promega) and Extracellular NanoLuc Inhibitor (Promega), BRET ratios (450 nm and 610 nm) were determined using a PHERAstar FSX plate reader (BMG Labtech). The preparation of SGC-590001 is described previously (Vasta *et al*, 2018). For nanoBRET experiments in HEKBlue cells, cells were transfected with NanoLuc-RIPK2 WT and stimulated with 1 ng/ml L18-MDP for 1 h prior to the inhibitor addition. Cell density was adjusted to $2 \times 10^5$ cells/ml, and cells were incubated with commercial nanoBRET tracer 6 (Promega) and various inhibitor concentrations for 2 h at 37°C. After that, NanoLuc

substrate mix (Promega) was added and BRET ratios were determined using a Victor3V plate reader (PerkinElmer, Waltham, MA). Residence time measurements were performed similarly, except cells were first incubated with the inhibitors, washed carefully to remove un-bound inhibitor, and then incubated with nanoBRET tracer and NanoLuc substrate. Emission signals were measured 60 times at 2-min intervals.

### Docking experiments and modeling

Ligand docking of CSLP37 and CSLP43 was performed using Auto-Dock molecular docking software (version 4.2.6) (http://autodock. scripps.edu/) with a standard protocol. The co-crystal structure of CSLP18 bound to RIPK2 was used for the docking experiment where CSLP18 was extracted from the RIPK2 using Discovery Studio 2016 Client software (http://accelrys.com/). CSLP37 and CSLP43 ligands were created and subjected to energy minimization using MM2 force field. Grid maps were selected based on CSLP18-RIPK2 co-crystal structure, and docking experiments were performed. The final pose was selected based on the lowest binding energy and analyzed using the PyMOL Molecular Graphics System, Version 2.0 Schrödinger, LLC (https://pymol.org/2/). Molecular surfaces were generated by mapping electrostatic charge on the models of RIPK2 (PDB: 5AR2) and XIAP (XIAP, PDB:1CQ9) using APBS (Jurrus *et al*, 2018) as implemented in PyMOL and are contoured between $-10$ and $+10$ kT/e as indicated in the figures.

### Data availability

The accession number for the coordinates and structure factors for the RIPK2 kinase domain bound by CSLP18 is PDB: 6FU5. The mass spectrometry proteomics data have been deposited to the ProteomeXchange Consortium (http://proteomecentral.proteomexchange. org) via the PRIDE partner repository (Vizcaino *et al*, 2013) with the dataset identifier PXD009724.

### Statistical analysis

Statistical analysis and calculation of $IC_{50}$ values were performed using Prism 6 (GraphPad Software). Unpaired Student's *t*-test and two-way ANOVA were used to determine statistical significance as appropriate.

**Expanded View** for this article is available online.

### Acknowledgements

This work was supported by the Ludwig Institute for Cancer Research Ltd. The SGC is a registered charity (number 1097737) that receives funds from AbbVie, Bayer Pharma AG, Boehringer Ingelheim, Canada Foundation for Innovation, Eshelman Institute for Innovation, Genome Canada, Innovative Medicines Initiative (EU/EFPIA) [ULTRA-DD grant no. ULTRA-DD 115766], Janssen, Merck KGaA, MSD, Novartis Pharma AG, Ontario Ministry of Economic Development and Innovation, Pfizer, São Paulo Research Foundation-FAPESP, Takeda, and Wellcome [106169/ZZ14/Z]. J.A.W. and K.V.M.H. are grateful for funding by Myeloma UK. G.D.C is supported by American Heart Association grant 15GRNT22970025. P.F. and S.P. are supported by a Wellcome Career Development Fellowship (095751/Z/11/Z) and the SGC. M.H. is supported by the Czech Science Foundation, grant number 18-24070Y. A.D. is supported by NIH grants R01CA190542 and R21AI124049. M.G-H is supported by a Wellcome Trust Fellowship (102894/Z/13/Z), a Sapere Aude: Danish Council for independent Research Starting Grant., and the EMBO Young Investigator Programme. We thank TetraLogic Pharmaceuticals for Compound A, Diamond Light Source for beamtime (proposal mx15433), the staff of beamline I04 for assistance with crystal testing and data collection, and members of our groups for helpful suggestions and discussions.

### Author contributions

Conceptualization, MG-H, AD, ANB, GDC; Investigation, MH, LS, BD, DMP, JCB, SP, JAW, CR, CS, SN; Methodology, KVMH, PF; Writing—Original Draft, MG-H, AD, MH, LS; Writing, MG-H, AD, MH, LS, GDC; Funding Acquisition, MG-H, AD, ANB, KVMH, GDC.

### Conflict of interest

The authors declare that they have no conflict of interest.

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
