## [Review Process File · The EMBO Journal]

Small-molecule inhibitors reveal an indispensable scaffolding role of RIPK2 in NOD2 signaling

Matous Hrdinka, Lisa Schlicher, Bing Dai, Daniel M. Pinkas, Joshua C. Bufton, Sarah Picaud, Jennifer A. Ward, Catherine Rogers, Chalada Suebsuwong, Sameer Nikhar, Gregory D. Cuny, Kilian V. M. Huber, Panagis Filippakopoulos, Alex N. Bullock, Alexei Degterev and Mads Gyrd-Hansen.

Review timeline:

Submission date:	3 rd March 2018
Editorial Decision:	29 th March 2018
Revision received:	7 th May 2018
Editorial Decision:	11 th June 2018
Revision received:	17 th June 2018
Accepted:	22 nd June 2018

Editor: Elisabetta Argenzio

Transaction Report:

1st Editorial Decision

29th March 2018

Thank you for submitting your manuscript on the characterization of small RIPK2 inhibitors that modulate RIPK2 function in a RIPK2 kinase activity-independent manner. The manuscript has now been reviewed by three expert referees whose comments are provided below.

As you can see, referees #1 and 3 find the findings novel and of high interest to the field and provide constructive feedback on how to further revise your manuscript prior to publication. Referee #2 is less supportive and only offers brief comments on the study. I have looked at the comments carefully and I agree with referees #1 and 3 that the analysis adds important new insight. The referees bring up some issues that should be resolved in a revised version. In particular, they point out that the clarity of the manuscript will greatly increase if you would edit and streamline the text incorporating their suggestions. Particular attention should be given to provide key information for non-expert readers to understand your experiments and the significance thereof. Given the overall interest of your study, I would like to invite you to revise the manuscript in response to the referees' reports.

REFeree REPORTS

Referee #1:

In this paper, Hrdinka et al. explore the mechanisms underlying the mode of action of RIPK2 inhibitors. The authors show that, surprisingly, RIPK2 kinase activity is dispensable for NOD2

signaling. Instead they convincingly demonstrate that the inhibitory mechanism relies on the disruption of XIAP binding to RIPK2 and the subsequent ubiquitination of the latter. Targeting RIPK2 might be of interest in a range of inflammatory conditions, such as multiple sclerosis and Crohn's disease. Therefore, the presented findings have important therapeutic implications and further might help to develop novel inhibitors to modulate RIPK2 signaling.

This is a well performed study that provides several interesting observations that are of interest for the wider audience of the journal. The results are straightforward and support the conclusions that are made. I have only some minor concerns that should be addressed:

- Are the novel RIPK2 inhibitors described here specific for RIPK2, as opposed to RIPK1/RIPK3?
- Can the authors show that the inhibitors are not cytotoxic.
- Why do the authors only test RIPK2-XIAP binding. What about cIAP1/2? Nachbur et al. (2015) have previously shown that WEHI-345 interferes with RIPK2-cIAP1 interaction. In fact, this should be acknowledged in the paper.
- Figure 5C: what is the extra lower band in the anti-GST blot that is missing in similar blots in Figures 5B and 5E.
- Figure 5E: Is it pull-down of recombinant proteins in vitro or with the use of cell lysate? Is the "Lysate" labeling correct?
- The manuscript sometimes lacks details necessary for the general audience (not experts in the field) to understand the experiments performed. Figures should be labeled better, for example it is not always clear which cells are used without checking figure legends.
- Proof reading for some typos is necessary, i.e. p3 first paragraph " granulatomous pathologies"; p5 second paragraph "ponatinb"

Additional suggestions that may improve the study:

- Testing new inhibitors in a relevant mouse model of inflammatory disease, where RIPK2 is implicated, would further validate the therapeutic value of the findings.

Referee #2:

Manuscript EMBOJ-2018-99372

Small-molecule inhibitors reveal an indispensable scaffolding role of RIPK2 in NOD2 signaling

By Hrdinka et al.

The authors investigated the effects of small-molecule inhibitors of RIPK2. They report that RIPK2 kinase activity is responsible for NOD2 inflammatory signaling. They went on to show that RIPK2 inhibitors function by neutralizing XIAP binding and XIAP-mediated ubiquitination of RIPK2.

Major points:

The manuscript is not within the primary focus of EMBO Journal and is therefore considered to be suitable for a more specialized journal on chemical compounds.

Referee #3:

The manuscript by Hrdinka and coworkers is of considerable interest as it provides a very thorough analysis of the mechanism by which small molecule compounds that bind to the ATP-binding pocket of RIPK2 can be used to manipulate RIPK2 function even though the kinase activity is not essential. This discovery is of considerable importance because RIPK2 is a key player in inflammatory signalling and these compounds have the potential to improve several pathologies. This is because the compounds disrupt interaction of RIPK2 with the E3 ligase XIAP, an essential step in assemble of a stable inflammatory signalling complex. It is likely that the compounds that target RIPK2 will be more selective and effective than compounds that target XIAP.

I have no major concerns, the manuscript is comprehensive and describes an elegant set of

experiments, is well-written and a pleasure to read.

Minor

- 1) It would be good to describe the classes of kinase inhibitors at the outset. Different classes are referred to in the introduction (and throughout) but for those not in the field it would be helpful to know about the significance of the classes earlier.
- 2) Please indicate r^2 in figure 3d.
- 3) The resolution of the structure is only 3.2Å. Because the details of the active site are important it would be helpful to show the electron density for this region and comment on the quality of the map in the main text.
- 4) Although the compounds will be described in detail elsewhere it would be helpful to include a simple schematic alongside the crystal structure in Figure 3.

1st Revision - authors' response

7th May 2018

Point-to-point response to referee comments.

Referee #1:

- Are the novel RIPK2 inhibitors described here specific for RIPK2, as opposed to RIPK1/RIPK3? The reviewer raises an important point here. We have now performed kinase assays with recombinant RIPK1 and RIPK3 and observed no inhibition of either kinase by CSLP37 and CSLP43 at concentrations up to 1 μ M. The new data is shown in Figure EV2C.

- Can the authors show that the inhibitors are not cytotoxic. We have included cytotoxicity assays with the CSLP37 or CSLP43 inhibitors and have not observed any toxicity of either in the cell types used in the study (HEKBlue, RAW264.7, U2OS, THP1 cells) treated with the compounds at concentrations up to 1 μ M. The data is shown in Figure EV2B.

- Why do the authors only test RIPK2-XIAP binding. What about cIAP1/2? Nachbur et al. (2015) have previously shown that WEHI-345 interferes with RIPK2-cIAP1 interaction. In fact, this should be acknowledged in the paper.

We focussed on characterising the RIPK2-XIAP binding because XIAP is indispensable for NOD2 signalling whereas cIAP1/2 both are not needed (Stafford et al. 2018, Damgaard et al. 2013). That said, we fully agree that the previous work on WEHI-345 and its effect on IAP-RIPK2 interaction should be included in the paper. Also, since cIAPs do ubiquitinate RIPK2 (at least in the absence of XIAP), we acknowledge that it is relevant to know if the inhibitors affect the interaction of RIPK2 with cIAPs. We have therefore performed interaction studies also with a GST-tagged BIR2 domain of cIAP1. The data is shown in Figure EV4B. The pulldown experiments show (as expected) that cIAP1-BIR2 can pulldown RIPK2 from cell lysates and that CSLP37 and CSLP43 both antagonise the interaction. These data suggest that XIAP and cIAP1 interact with RIPK2 via a similar mechanism. The experiment is described on page 10 and we have included a discussion of the cIAP1-RIPK2 interaction on page 14/15.

- Figure 5C: what is the extra lower band in the anti-GST blot that is missing in similar blots in Figures 5B and 5E.

The lower band is present also in Figures 5B and 5E although it is weaker in those blots due to lower exposure of the blots. We have performed anti-GST blots on the purified GST-XIAP/cIAP1-BIR2 preparations and in these we observe a band of approx. 26 kDa in addition to a band of the expected size of the recombinant protein (see figure to the right). This is most likely a cleavage fragment of the fusion protein, which we believe is the same signal detected in Figure 5B-5D. We have indicated the band with an asterisk and explain the nature of the detected signal in the accompanying figure legend.

- Figure 5E: Is it pull-down of recombinant proteins in vitro or with the use of cell lysate? Is the "Lysate" labeling correct?

This is indeed a pulldown of recombinant proteins and we have corrected the labelling to "Input" instead of "Lysate".

- The manuscript sometimes lacks details necessary for the general audience (not experts in the field) to understand the experiments performed. Figures should be labeled better, for example it is not always clear which cells are used without checking figure legends.

We appreciate the reviewer's comment and agree that the results section in some places lacked sufficient detail about the described experiments. We have included additional information about the experiments shown in figures throughout (pages 4-11). We have also indicated the cell line used in all figure panels where relevant.

- Proof reading for some typos is necessary, i.e. p3 first paragraph "granulatomous pathologies"; p5 second paragraph "ponatinb"

We have proof-read the revised manuscript carefully and any found typos have been corrected.

Additional suggestions that may improve the study:

- Testing new inhibitors in a relevant mouse model of inflammatory disease, where RIPK2 is implicated, would further validate the therapeutic value of the findings.

This is a good suggestion and will be important to further validate the therapeutic value of the CSLP compounds. However, we feel it is beyond the scope of the current study in which we utilise the CLSP compounds primarily to understand the mechanism of action of RIPK2 inhibitory compounds rather than to assess their therapeutic value.

Referee #2:

Major points:

The manuscript is not within the primary focus of EMBO Journal and is therefore considered to be suitable for a more specialized journal on chemical compounds.

We respectfully disagree with the reviewer. In the study we utilise chemical inhibitors as tools to elucidate fundamental mechanisms for how RIPK2 (and XIAP) facilitate NOD2 signalling. As such, the scope of the study is not solely to characterise chemical compounds but rather to uncover the molecular mechanisms controlling cellular signalling.

Referee #3:

Minor

1) It would be good to describe the classes of kinase inhibitors at the outset. Different classes are referred to in the introduction (and throughout) but for those not in the field it would be helpful to know about the significance of the classes earlier.

We appreciate reviewers suggestion and added a brief description at the outset of our discussion of inhibitors on page 4: "Small molecule kinase inhibitors are categorized into multiple classes, depending on their mode of binding (Roskovski, 2016). This includes type I inhibitors that interact exclusively within the ATP binding pocket, type II inhibitors that bind both to the ATP and an additional back-pocket created when the activation segment of a kinase adopts an inactive conformation, and type III molecules that bind exclusively to this allosteric back pocket. Curiously, we observed that a subset of known RIPK2 inhibitors belonging to different classes displayed potent (nanomolar) cellular activities, including ponatinib (a type II inhibitor) and GSK583 (an ATP-competitive type I inhibitor), and that these molecules also antagonized NOD2-mediated ubiquitination of RIPK2 (Figure 1C; Figure EV1A) (Canning et al., 2015). This implied that the kinase activity of RIPK2 is required for its ubiquitination and, thus, for NOD2 responses."

2) Please indicate r2 in figure 3d.

We have now indicated R1-R3 in Fig 3D and 3E to make the panels more accessible to the reader.

3) The resolution of the structure is only 3.2Å. Because the details of the active site are important it would be helpful to show the electron density for this region and comment on the quality of the map in the main text.

This is a good point raised by the reviewer. We have included the electron densities in Figure EV2E and have included a short comment in the text (page 8): "The structure is at 3.2 Å resolution, and the electron density map is of sufficient quality in the region of the inhibitor to place the inhibitor and its relevant functional groups with reasonably good precision (Figure EV2E)."

4) Although the compounds will be described in detail elsewhere it would be helpful to include a simple schematic alongside the crystal structure in Figure 3.

We have included a schematic of CSLP18, CSLP37, CSLP43, CSLP48, and CSLP55 in Figure 3C.

2nd Editorial Decision

11th June 2018

Thank you for submitting a revised version of your manuscript. It has now been seen by two original referees whose comments are shown below.

As you will see they both find that all criticisms have been sufficiently addressed and recommend the manuscript for publication. However, before we can officially accept the manuscript there are a few editorial issues concerning text and figures that I need you to address in a final revision.

Thank you again for giving us the chance to consider your manuscript for The EMBO Journal, I look forward to your revision.

REFeree REPORTS

Referee #1:

All my concerns were well addressed. I have no further comments.

Referee #3:

All my concerns have been addressed and in my view the manuscript is suitable for publication.

YOU MUST COMPLETE ALL CELLS WITH A PINK BACKGROUND ↓
PLEASE NOTE THAT THIS CHECKLIST WILL BE PUBLISHED ALONGSIDE YOUR PAPER

Corresponding Author Name: Gyrð-Hansen
Journal Submitted to: The EMBO Journal
Manuscript Number: EMBOJ-2018-99372